# Smart Learning Environments during Pandemic

Melissa Rutendo Mutizwa [1] 🆔, Fezile Ozdamli [2],* 🆔 and Damla Karagozlu [3]

1   Department of Computer Information Systems, Near East University, 99138 Nicosia, Cyprus
2   Department of Management Information Systems, Near East University, 99138 Nicosia, Cyprus
3   Department of Management Information Systems, Cyprus International University, 99258 Nicosia, Cyprus
*   Correspondence: fezile.ozdamli@neu.edu.tr

**Abstract:** Education was one of the many day-to-day activities affected by the novel coronavirus pandemic (COVID-19). When countries began to shut down in April 2020, nationwide lockdowns, self-isolation, or quarantine became the new normal for everyone. The education sector was kept alive by smart learning environments. Now, more than ever, online learning and tools were implemented. This study aims to systematically review the literature on the impact of the pandemic on smart learning environments. The method adopted in this paper is a systematic literature review, and it will use the PRISMA technique. A qualitative approach was applied in the data collection process to achieve the aim. The essential advantage was that smart learning environments were convenient and easily adapted by students during the pandemic. The main challenge was connectivity issues and failure to adapt to non-traditional methods. The paper concluded a rise in the usage of smart learning environments, and educators and students adapted quickly to the shift.

**Keywords:** smart learning; COVID-19 pandemic; e-learning; learning management system

## 1. Introduction

Smart learning has been a trendy term for education in today's digital age. According to [1], smart learning shows how modern technologies make it easy for learners and educators to digest knowledge and skills in a well-organized and competent way and more conveniently. Smart learning includes educational factors in which the importance is concentrated on the student's technology use. For smart learning to be effective, the main requirement is for the students to know how the technology works since it depends on the hardware and software aspects and how they are segmented in the classes or the online training [2]. Some advantages of smart learning are that it helps to kindle interest in education by introducing participants to on-demand learning with the help of videos, online web conferences, and it also ensures that education reaches every student and improves student–teacher interaction [3]. Smart learning environments are, therefore, physical environments for learning enhanced with digital devices whose aim is to improve and accelerate training that supports the technology. According to [4], a smart learning environment includes technology, students, instructors, or an instructional system, the settings in which learning occurs, the support staff including designers and technical specialists, and the class's culture, course, institution, and community. In the year, 2020, there was a notable jump in smart learning environment tools because of how the pandemic disrupted the routine the world was used to. The COVID-19 pandemic began suddenly in Wuhan, China, in December 2019, and abruptly spread to more than 200 countries on five continents. Information obtained [5] as of 11 December 2020 shows that there had been a total of 1,576,974 deaths worldwide and 48,053,025 cases that tested positive but have recovered and been discharged. However, it should be noted these numbers change every day. The briskness of how the pandemic spread just by something as mundane as a bug presented governments and school systems with difficult choices. They understood it was hard for families and businesses to operate safely without endangering people's

lives and so they had to devise ways on how to keep people's lives safe while protecting their livelihoods. Most governments immediately gave an announcement to all people, including students, to remain home for quarantine until further notice. As a result, students woke up one day to travel restrictions, compulsory lockdown, and curfews and could not return to their schools and universities, [6,7] stated that over 1.5 billion learners worldwide could not attend a school or university due to the COVID-19 virus. In 2020, worldwide, the education sector faced a significant impact as a result of COVID-19, with universities currently obliged to shift their teaching to smart learning. Most institutions adopted some of the following video conferencing tools to gain momentum Google Meet (New York City, NY, USA), CISCO (San Jose, CA, USA), WebEx (San Jose, CA, USA), Zoom (Zoom Video Communications, Inc., San Jose, CA, USA), Microsoft Teams, and learning management systems such as Blackboard (Version 5.11.1), Edmodo (Version 10.32.0), and Moodle (Version 3.11)

Therefore, this study seeks to determine how the pandemic impacted the usage of these smart learning environments during the 2020 academic year. This study will review published scholarly articles from different authors and their views about smart learning environments and extract necessary data.

### The Aim of the Study

The aim is to assess the impact of COVID-19 on smart learning environments and after a thorough evaluation and research to give a clear definitive answer to the research questions stated below

- RQ1: What are the benefits of smart learning environments during the COVID-19 pandemic?
- RQ2: What are the challenges in smart learning environments caused by the pandemic from students, educators, and educational institutions?
- RQ3: From the e-learning tools mentioned in this paper, which tools were the most preferred tools for smart learning environments during the pandemic?

## 2. Materials and Methods

### 2.1. Research Setting

A systematic literature review will be used to conduct this study. According to [8,9] it is a step-by-step thorough and careful review of research results. He also states that its aim is not to pile all the available evidence on a research question, but it aims to support the development of evidence-based guidelines.

This methodology will be using the PRISMA technique for systematic reviews since it is instrumental in obtaining quality evidence for use in this paper. It will map out the number of records identified, included, and excluded, and the reasons for exclusions; this will be explained in detail in the next section of the paper. An advantage of using PRISMA is that it reveals the review's quality by allowing readers to assess strengths and weaknesses.

### 2.2. Data Collection

Published papers that focused on smart learning environments concerning the pandemic were searched and obtained. These articles were electronically obtained by searching well-known and online databases of the Near East University Grand Library Electronic Resources. The articles for review in this paper were obtained from Science Direct, the Web of Science, springer link, and Scopus. Keywords used to find the articles on the subject, were used; ("Pandemic" OR "COVID-19" OR "Corona Virus") AND ("Smart Learning Environments" OR "LMS" OR "Learning Management Systems" OR "E-Learning" OR "Distance Education" OR "Online Learning"). After using these keywords in selected databases, many articles were ready to be used. However, not all of them were relevant to the topic. The search was narrowed using the following criteria in Table 1.

**Table 1.** Article selection criteria.

| Inclusion | Exclusion |
| --- | --- |
| • Articles that focus on the COVID-19 pandemic<br>• Open access papers<br>• Articles<br>• Articles in English | • Non-English articles<br>• Ones which were not open access |

### 2.3. Article Selection Criteria

Four phases were determined to meet the paper's inclusion criteria. The first step was the initial search using the above-mentioned keywords; the years were filtered to 2020. Since this article is related to COVID-19, only articles written and published in 2020 were selected. The language was filtered to English so that time would not be wasted looking for translations. Finally, the document type was narrowed down to only published articles and research papers. The summary of the database search is shown below:

The first database searched was EBSCO; the initial search resulted in 3500 articles, but after selecting the year range to be 2020 and the article type as research articles, and the language as English, 500 articles remained. The result was ten articles on the Web of Science after using the same keywords, selecting document type as article and article being open access, the remaining was 2 articles. Scopus came up with 21 articles, but after invoking our inclusion criteria, the remaining documents were 13. Springer brought up 182; after the screening, the remainder was 18 articles. Moreover, Science Direct had 2483 articles at first, but after the screening, the remaining 300 articles. So that brought the total to 833 articles.

### 2.4. Searching Process

The following PRISMA FLOW diagram summarizes the data collection process and how it was further filtered to remain relevant. PRISMA diagram is a simple and realistic solution for showing the flow of studies [10]. As seen in the diagram, Figure 1, of the 833 articles that remained after the inclusion/exclusion criteria, further screening was performed as follows. Some papers in the databases only offered the abstract, and the full text was not available; these papers were all excluded, and the ones that included any other language that is not English in total 433 articles were removed. From the remaining 400 papers, 100 papers from different databases were similar in title or context, so they were excluded, and there remained 300 papers. From the 300 papers, the authors went through the abstract to make sure they were relevant to this paper's topic, and 200 articles remained. Since this paper focuses on smart learning, only documents in line with education and academics were selected from the 200, and the remaining 45.

### 2.5. Findings

This paper reviews the past literature to determine how the pandemic influenced smart learning environments to answer the research questions. The Table 2 below summarizes the studies from each of the 45 papers used in this research.

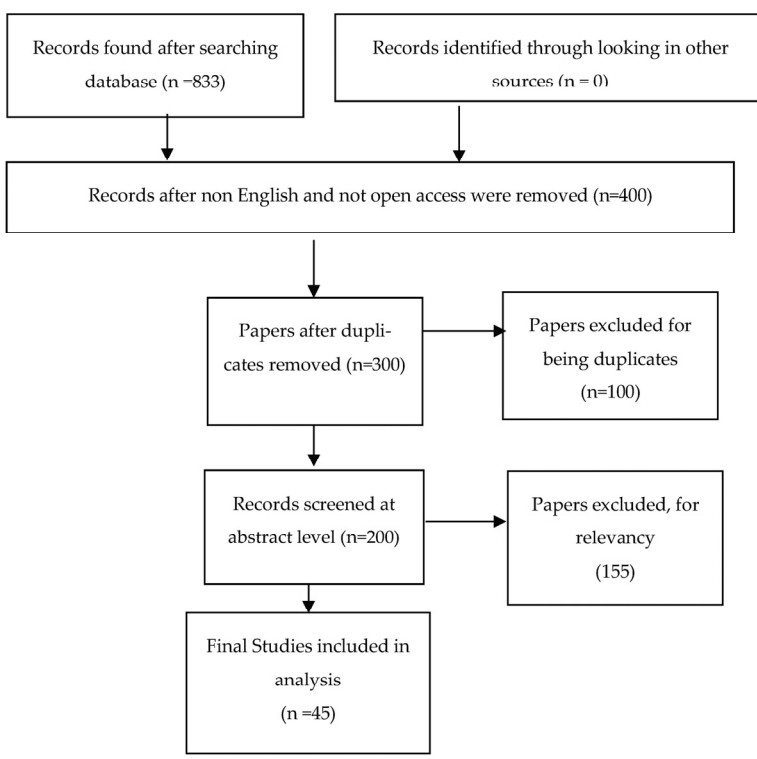

**Figure 1.** PRISMA flow diagram of the publication selection process.

**Table 2.** Analyzed information of the related studies.

| Author and Year | Aim | Method | Result |
|---|---|---|---|
| [11] | Evaluation of the distance education challenges experienced by students during the pandemic process by families. | Qualitative | Families stated that they have a responsibility to help students in the distance education process. Many parents stated that they faced obstacles in helping their children with distance education during the pandemic. |
| [12] | Identifying the difference made by the Moodle environment for learning English. | Quantitative and qualitative | Students satisfied with using Moodle to support English learning. |
| [13] | To evaluate if hybrid learning will continue after COVID. | Qualitative | Online educational methods need to be incorporated in the current training programs. |
| [14] | The impact of the pandemic in the cloud computing environment. | Qualitative | Dependence on cloud computing and other technologies has increased due to the pandemic. |
| [15] | Examine teaching and learning enhancement aspects of open-source courses. | Quantitative | MOOCs have been shown to have a significant direct impact on higher education as they improve educational outcomes. |
| [16] | Offering different digital procedures for medical students to perform better and achievement. | Qualitative | Results show that students are satisfied with the transition to a collaborative e-learning environment and its procedures. |

**Table 2.** *Cont.*

| Author and Year | Aim | Method | Result |
|---|---|---|---|
| [17] | Evaluating students' experiences and satisfaction with e-learning applications during the pandemic process. | Quantitative | Results showed that students are delighted with Google Hangouts for resource distribution and Google Classroom and Moodle for course management. |
| [18] | To investigate the effect of remote teaching on pharmacy education and provide recommendations. | Quantitative | The results showed that the lack of interaction in online education is a negative factor in student satisfaction. In addition, emergency distance learning practices caused teachers to use alternative assessment methods. |
| [19] | Investigating the use of social media in education. | Quantitative | Using social media for learning English increased dramatically during pandemic COVID-19 participants showed a positive attitude towards social media use. |
| [20] | Determining the effect of e-learning applications on institutions. | Quantitative | Teachers say it is challenging to do course objectives due to the lack of practical lab work this leads to which leads to dissatisfaction with online courses when compared with on campus education. |
| [21] | To make a survey of students in the various universities regarding e-learning systems implemented during the COVID-19 outbreak. | Quantitative | E-learning positively impact students and becomes a substitute learning process for lecturers and students. |
| [22] | Determine students' views on e-learning applications in medical physics education. | Qualitative | The majority of students found that watching videos of pre-recorded lectures and Practical sessions as well as answering short questions was helpful, but it was difficult to focus at home due to distraction and poor Internet connection. |
| [23] | To understand the motivation for developing writing and speaking skills in language education and its impact on using language learning technology. | Quantitative | Students motivated by asynchronous online collaborative writing were more likely to enjoy online learning. |
| [24] | Determining students' understanding of face-to-face and online education during the COVID-19 process. | Quantitative | The students participating in the study were positive towards online education, but were not sure if the quality of online education matched the traditional setup. |
| [25] | To address the best practices for teaching cytology and pathology remotely. | Qualitative | Learners expressed their willingness to attend online classes in the future, even if the traditional classroom learning option is available. |
| [26] | Evaluation of nursing faculty student' use of distance education applications. | Qualitative | A theory-guided, caring approach supports the needs of both faculty and nursing students. |
| [27] | Explanation of how to plan a virtual site visit for nursing accreditation. | Qualitative | Educators can be tested for compliance with virtual accreditation visits. |

**Table 2.** *Cont.*

| Author and Year | Aim | Method | Result |
|---|---|---|---|
| [28] | Evaluate online classroom practices and determine whether they can aid medical learning during the lockdown process. | Quantitative | Most students mostly favored online classes. Those who did not favor online classes said poor internet connection was the main hindrance. |
| [29] | How does the internet help to transform the mathematics classroom and mathematics teacher education? | Qualitative | Modern students want a say in how they are taught and what they are taught. |
| [30] | Determining the results of organizational adaptation to artificial intelligence strategies. | Qualitative | Findings conclude that AI adaptation processes need to include the following empathy and original, representation, fundamental needs, and motivations. |
| [31] | Determining student satisfaction levels towards the learning process realized by video conference. | Quantitative | The majority agreed that the sessions were intellectually challenging, but the instructors were dynamic, and encouraged students to participate. |
| [32] | To determine which of the strategies presented have the potential to benefit FLHCP that are concurrently enrolled in the university. | Qualitative | Hundreds of mental health online resources were identified; however, less than a quarter were either developed based on evidence or empirically evaluated. |
| [33] | Evaluating the use of the WhatsApp application in the online evaluation process. | Questionnaire | WhatsApp was the most comfortable mode of communication and was used as a way to conduct theory and practical exams. |
| [34] | Assessing the possibility of using open educational resources as alternatives to tackle challenges. | Qualitative | Various initiatives have been taken to provide open and online education. |
| [35] | Identifying factors affecting instructors' intention to reuse the LMS. | Qualitative | The most popular challenges instructors faced during their Experience with Google Classroom were inadequate internet service and students lack of interest. Service quality also had no positive influence on instructor satisfaction. |
| [36] | Designing functional robots for older people during the COVID-19 pandemic. | Qualitative | The developed social robots provide a promising alternative to address social isolation and loneliness during the COVID-19 pandemic. |
| [37] | Examining student's effectiveness in online nursing leadership education. | Qualitative | Results show student-created questions to be a useful learning tool. |
| [38] | To assess the impact of lockdown on undergraduate and postgraduate learners. | Quantitative | Teachers used many platforms for teaching and evaluation. Learners suffered from stress depression anxiety because of the lockdown. Some students who were from remote and marginalized areas did not have access to reliable internet. |
| [39] | To evaluate the LMS acceptance levels of medical students. | Qualitative | Increasing informative activities on LMS increases the intention to use distance education systems to improve students' acceptance level. |

**Table 2.** *Cont.*

| Author and Year | Aim | Method | Result |
|---|---|---|---|
| [40] | Explain how the education system has changed. | Qualitative | Increase in Zoom downloads. Both students and instructors, to a certain extent, face challenges in virtual learning. |
| [41] | Determining the necessary foundations of e-learning carried out during the pandemic process. | Quantitative | University's initiative for online teaching and learning succeeded through following instructions. |
| [42] | To evaluate the use of online learning applications. | Qualitative | Adoption of distance education, blended learning e-learning, and course management system is the appropriate response to the pandemic. |
| [43] | Determining the changes in teachers' perceptions in the e-learning process. | Quantitative | Educators worked to create opportunities for interaction and provide learning experiences in an online Environment. |
| [44] | Offering alternative ways to help students in the distance education process to overcome their difficulties. | Quantitative | The results show that the digital divide is an obstacle for students in the e-learning process. |
| [45] | To portray the online learning barriers students, face and their alternatives to cope with them. | Quantitative and qualitative | According to the results, students encountered eyestrain, familiarity with e-learning and slow internet connection during the distance education process. |
| [46] | Exploring distance learning at Bezmialem Vakıf University. | Quantitative | The hybrid models, asynchronous and synchronous were applied and were more beneficial in some subjects. However, it was not possible during distance education in practical and internship practices. |
| [47] | Disclosure of the perceived usability of Microsoft Teams. | Quantitative | Results show the similarity and equivalency be the Perceived Ease of Use factor of TAM having a more significant similarity with the system usability scale. |
| [48] | Highlighting how augmented reality improves and enhances its users. | Qualitative | The results show that the use of AR in medicine can change the way surgeries are performed. |
| [49] | Determining students' perceptions of adopting, using, and accepting online learning. | Quantitative and qualitative | attitude, motivation, self-efficacy, and use of technology play a significant role in the mental engagement and performance of students. |
| [50] | Evaluation of the effectiveness of e-learning applications by dental students. | Quantitative | Students were not satisfied with the learning management system and quality of learning resources available. |
| [51] | Determining how cloud-based systems can be used to achieve smart solutions that can be adjusted to low costs. | Qualitative | The Alexa smart speaker's initiative for university has aims to improve connections and data manipulation through smart cloud services. |

**Table 2.** *Cont.*

| Author and Year | Aim | Method | Result |
|---|---|---|---|
| [52] | To identify educators' behaviors related to the corona virus pandemic. | Quantitative and qualitative | That teacher attitudes toward change, and perceptions of administrative support was related with resilience and burnout they suffered because of the pandemic. |
| [53] | Analyze factors that predict the use of e-learning in sports science education during COVID-19. | Quantitative | Thee-learning environment and tools maximize the positive effects in the use of e-learning. The participants considered it to be user-friendly and enhance their emotions to the advantage of the instruments during COVID-19. |
| [54] | Evaluating reliable and cost-effective distance learning strategies and online tools. | Quantitative | It is stated that using Google Classroom as a learning tool and WhatsApp group for sending messages is effective. |
| [55] | Identifying factors that affect accepting YouTube as a learning tool. | Quantitative | Acceptance of YouTube as a learning tool was related to these factors perceived ease of use, perceived usefulness, social influence, individual and environmental factors, to guarantee students acceptance of online resources. |
| [56] | Evaluating whether e-learning works in traditional Chinese medicine teaching. | Quantitative | As a result, online education has been identified as a good alternative when it can no longer be done face to face. |

## 3. Results

### 3.1. Benefits of Smart Learning Environments

According to many authors [21,25,27], students agreed that e-learning is beneficial because it can be done anywhere and anytime using various learning management platforms. Students also agreed with learning methods and tools that e-learning can help the learning process to continue even amid a pandemic as long as they have the internet and the online classes are engaging their minds [37,53]. A paper based on a South African university also supported this claim [44], uses e-learning for teaching and e-learning university courses, and is a significantly effective way to ensure relationships between universities and the private sector. It also improves education and its outcomes [15].

Using social media as a tool to learn English was another advantage discussed in the literature. [19], stated that because it enabled students to be active and social media can be used for speaking and reading skills. Research has shown students are keen to continue utilizing smart learning environments even after the pandemic has passed [13,25]. During the lockdown process, individuals, especially young people, had more time to devote to themselves. In this process, smart learning environments have been useful for foreign language learning.

Learning and studying from home offer an adaptable environment for students. Students prefer pre-recorded lectures and viewing videos on YouTube of practical sessions [22]. It supports [42] the theory that students work according to the convenient schedule for them, as the lecture material will always be available. Some students also evaluated smart learning environments as user-friendly and considered the content instructive satisfying, positive and motivating, and constructive learning environments [16,17,31,39].

For asynchronous activities [23], written narrative or video discussions can be held at different locations and at different times. Educators can raise awareness about the topic by

creating short videos on the learning topic of the week. Such videos allow learners to see and hear the faculty as another way of caring presence and connection by using various web-based and smartphone-based applications to create the videos, or they may be created in the LMS [26].

Literature also supported combining LMS and virtual technology such as virtual site visits, artificial intelligence, augmented reality, and even social robots to engage students in smart learning environments [27,30,36,48,51].

Another advantage of smart learning environments is that they enable polling tools to foster active engagement. Smart learning platforms such as CISCO WebEx include such polling tools already. Polling tools can be used to record student participation automatically [22]. They can also be combined with student-generated questionnaires [37] because some modern students raised concerns about wanting to have a say in what they are taught when studying online [29].

### 3.2. Challenges on Smart Learning Environments Caused by the Pandemic

Some previous studies suggested that e-learning tools for online exams require learning more about the system in use, which can be complicated for some. The authors of [11,18,39] state that students who were used to instructive lesson-based learning said they had challenges managing the learning process. Students accustomed to teachers' didactic lessons faced the most difficulty because the immediate change from face-to-face to e-learning was fast. The research of [17,49] supported this by stating that even though some students were ok with the shift to complete smart learning environment learning, the complete shift was not preferred by some students. The justification being most students prefer the blended class, especially for courses that require laboratory work. The study of [46] agreed that distance education in practical and internship practices was not possible.

Most students who faced dissatisfaction with online learning during the pandemic were medical students [26] because in as much as smart learning offers flexibility and convenience, certain aspects such as hands-on practical and clinical experience could never be replaced for students studying programs such as nursing [22,37]. The study of [50] states that dental students were dissatisfied with LMS, quality of learning resources available, and teachers' training level for online lectures. In their study of medical students, [17] supported this theory because students preferred blended classes, especially for courses requiring a laboratory.

Some research suggests that synchronous modes are necessary for learners only due to the social nature of language learning [23,43]; however, in the asynchronous mode of learning, for example, in pre-recorded lectures and external resources such as YouTube [22], there is a lack of interaction between the teacher and student, and this causes great difficulty. This challenge has been addressed quite frequently in the literature. The authors of [56] stated that students are less interested in their teachers in the online education process and are more distracted by external factors in their environment. The study by [28] drew the same conclusions and said chores and distractions at home and poor internet connectivity affected online classes' smooth flow. Healthwise, continuous screen exposure caused eye strain and headaches, while some even suffered from stress and anxiety [32,49].

On the same note, students faced another challenge of technical issues such as poor network and video inconsistency due to the remoteness of the student's location. Students residing in rural and remote areas faced poor internet connectivity [38,40,41]. It was supported by [44], who brought to light the effect of the digital divide and how it limits most students, in particular, those in remote areas; some do not have reliable internet at home. Moreover, according to the study conducted by [28], students who did not favor online classes said poor internet connection was the main hindrance.

That is why many teachers have turned to alternative assessment methods such as written assignments [18]. Other faculty who used strategies to nurture presence and student engagement yielded improved student evaluation scores in the understanding of course material, interest, meeting learning objectives, increased knowledge, and overall

effectiveness of faculty. The authors of [26,35] stated that instructors agreed that the internet is a significant factor as not all students are in a position to afford to pay for an internet service provider.

### 3.3. Most Preferred Tools for Smart Learning Environments during the COVID-19 Pandemic

Google Cloud infrastructure proved to be popular because it has several tools for distance learning google meet, google forms for quizzes, and google drive for sharing files [41,54].

The students' opinion of the most effective smart learning tool according to [21] was Google Hangouts, which was preferred for course delivery, and Moodle as a medium of assessments. Google Classroom was the most widely used, according to [9], followed by the social network WhatsApp and then Edmodo. The authors of [33] conceded with this statement by concluding that WhatsApp was the most comfortable mode of communication, and it was used as a medium to conduct theory and practical exams. WhatsApp is the most used social media tool to spread information related to distance learning. The simplicity and speed factor made WhatsApp more widely used than other social media [54], and even though WhatsApp was a convenient medium for communication, educators faced a challenge in which open book exams turned out to be opportunities for cheating [33,44] argued that even though students preferred WhatsApp for communicating, it was not fully adopted by universities for e-learning.

Moodle was observed to positively affect student learning of the English language [12]; in their paper, the authors state that it increased motivation and productivity. It increased students' grades and was beneficial to students. The authors of [42], however, disagreed and their study results showed Moodle had a disadvantage in online communication with students. However, [12] argued that Moodle works better when integrated with face-to-face. Moreover, it works better when used in conjunction with YouTube videos being uploaded on Moodle [42]. A person using YouTube might perceive that the content of YouTube is useful for their learning and receive positive feedback from their social circle because videos help users improve their skills and cognitive ability by gaining knowledge. The research of [24,55], in their study results, showed that males prefer YouTube more than females.

Zoom experienced a considerable increase in downloads since the lockdown started [40]. The research by [54] stated that Zoom downloads increased from 170,000 in mid-February to 2.5 million in late March [16,42] and argued that Zoom was preferred to Moodle and Blackboard because it was considered easy and straightforward. Moodle was also most of the respondents in the study. The study by [38] used the Zoom app for attending online classes or e-lectures. Instructors found using Google classroom was an effortless task [35,45,54] supported this theory by saying students preferred it because of its simplicity.

## 4. Discussion and Conclusions

The dependence on smart learning environments and other technologies took a major increase due to the current situation of COVID-19 [14]. Moreover, this systematic literature review aimed to review previous literature on smart learning environments during the pandemic and evaluate students' and teachers' satisfaction and adaption to using them and see the most preferred tools according to previous studies. The essential advantage recognized was that smart learning environments were convenient and easily adapted by students during the pandemic, with the main challenge being connectivity issues and failure to adapt to non-traditional methods.

Seeing as the COVID-19 catastrophe is not going anywhere anytime soon, students and educators must learn to live and survive the present crisis as it is only the beginning, and they will need to adapt and find ways to overcome challenges faced using smart learning environments. Therefore, the education sector needs to update and procure software that will enable the continuance of teaching remotely and ensures it runs without interruption.

It is important to highlight that the results were mainly concerned with students' perspectives. Few papers addressed educators and institutions; therefore, future research

should provide insight into how universities adapted to a complete shift from the classroom setting to using hundred percent online using various LMS tools and how that affected the school curriculum in terms of enrolment, seminars, and graduation ceremonies.

It should be noted from the results that most papers were from Asia and the Middle East, with a few from North America. Only one paper [44] was from Africa, of which it probably faced the most challenges in the transition because of the digital divide. Future research must also include third-world countries as well.

Most studies were conducted during the school year when people were learning to live in a world where going outside was risky [22,34,39,45,46,52,53,55,56]. The study surveyed teachers during the 2019–2020 school year when the pandemic had just begun. Future research must also focus on post-COVID-19 environments and the 2020–2021 and 2021–2022 academic year, the new normal, and how smart learning environments usage will be affected when the world is entirely normal again, as only a few papers [13,25] addressed if distance learning should be said to be here to stay for good or not.

Other limitations were that only articles were searched based on specific keywords in the selection process, and the language is English. In the future, research should also consider conference papers and use another language besides English to get opinions of other cultures.

**Author Contributions:** Investigation, M.R.M., writing and original draft preparation, M.RM., writing of the manuscript M.R.M., Review, F.O. and D.K., editing, F.O. and D.K., supervision, F.O. and D.K. All authors have read and agreed to the published version of the manuscript.

**Funding:** This research received no external funding.

**Institutional Review Board Statement:** Not applicable.

**Informed Consent Statement:** Not applicable.

**Data Availability Statement:** Not applicable.

**Conflicts of Interest:** The authors declare no conflict of interest.

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
