# Peer review of "Smart Learning Environments during Pandemic"

_2813-4346, doi:10.3390/higheredu2010002_

Round 1
Reviewer 1 Report
The article provides a systematic review of the state of the art of intelligent learning during pandemics using the PRISMA model. The strengths of the article are the appropriate use and explanation of the PRISMA model. As weak points, I consider that having limited the selection of articles only to the English language may have negatively affected the results. However, this is a limitation that the authors note in the article. I consider that the article is at a very advanced stage of development and that, therefore, no changes or improvements are required.
The article is well written and its contribution is relevant.
Author Response
Thank you for your positive comments.
Reviewer 2 Report
This manuscript is not ready for evaluation as there is something seriously wrong with the bibliographic references. Perhaps the bibliography has been sorted/re-ordered and the references not updated? Or maybe the references from a different paper have been used and not changed?
Starting from the very beginning:
"According to [1], smart learning shows how modern technologies make it easy for learners and educators to digest knowledge and skills in a well-organized and competent way and more conveniently."
[1] does not talk about that.
"For Smart learning to be effective, the main requirement is for the students to know how the technology works since it depends on the hardware and software aspect and how they are segmented in the classes or the online training [2]".
[2] does not talk about that.
"Some advantages of smart learning are that it helps to kindle interest in education by introducing participants to on-demand learning with the help of videos, online web conferences, and it also ensures that education reaches every student and improves student-teacher interaction [3]"
[3] does not talk about that.
"According to [4], a smart learning environment includes technology, students, instructors, or an instructional system, the settings in which learning occurs, the support staff including designers and technical specialists, and the class's culture, course, institution, and community"
[4] does not talk about that.
A systematic literature review will be used to conduct this study. According to [7], it is a step by step thorough and careful review of research results.
[7] does not talk about that.
I stop here, as I have already invested too much time on accessing and reading other papers; time that I should be devoting to studying the paper under review.
I believe this is something that should be done by the authors, not the reviewers. I will be happy to do the review once the authors have had a chance to go through their manuscript and make sure everything is the way they meant it to be.
Author Response
First of all, thank you for your constructive comments. Reference errors made while editing the format have been corrected.
Round 2
Reviewer 2 Report
The paper is a *perfect* match for this journal, as it provides a detailed review and discussion on a recent trend in higher education, that of smart learning environments during the pandemic.
The methodology followed is suitable. The search terms and the inclusion/exclusion criteria are broad enough to include a sufficient volume of research and narrow enough to remain focused. Overall it is a very well designed and executed review work, I send my congratulations to the authors. I have no serious changes to recommend, and this does not happen that often!
One minor issue is that Figure 1 is not legible in the manuscript that was provided for the review. It possible that there has been some problem with the track changes feature and the way it was transformed to PDF. Unfortunately there is no way for me to see what was meant to be there. PRISMA is quite standard, so I have no reason to doubt that the authors will provide a valid diagram for their finalized manuscript. But I do have to note that the figure needs to be checked before being published.